# Terpenes and Terpenoids in Plants: Interactions with Environment and Insects

**DOI:** 10.3390/ijms21197382

**Published:** 2020-10-06

**Authors:** Delbert Almerick T. Boncan, Stacey S.K. Tsang, Chade Li, Ivy H.T. Lee, Hon-Ming Lam, Ting-Fung Chan, Jerome H.L. Hui

**Affiliations:** 1School of Life Sciences, The Chinese University of Hong Kong, Shatin, Hong Kong; delboncan@gmail.com; 2Center for Soybean Research of the State Key Laboratory of Agrobiotechnology, The Chinese University of Hong Kong, Shatin, Hong Kong; 3Simon F.S. Li Marine Science Laboratory, The Chinese University of Hong Kong, Shatin, Hong Kong; stacey.tsang@link.cuhk.edu.hk (S.S.K.T.); R21381184@link.cuhk.edu.hk (C.L.); ivyleeting@yahoo.com.hk (I.H.T.L.)

**Keywords:** Terpenes, terpenoids, plants, insect, abiotic, biotic, interaction, environment

## Abstract

The interactions of plants with environment and insects are bi-directional and dynamic. Consequently, a myriad of mechanisms has evolved to engage organisms in different types of interactions. These interactions can be mediated by allelochemicals known as volatile organic compounds (VOCs) which include volatile terpenes (VTs). The emission of VTs provides a way for plants to communicate with the environment, including neighboring plants, beneficiaries (e.g., pollinators, seed dispersers), predators, parasitoids, and herbivores, by sending enticing or deterring signals. Understanding terpenoid distribution, biogenesis, and function provides an opportunity for the design and implementation of effective and efficient environmental calamity and pest management strategies. This review provides an overview of plant–environment and plant–insect interactions in the context of terpenes and terpenoids as important chemical mediators of these abiotic and biotic interactions.

## 1. Plant Terpenes in Abiotic Stress

### 1.1. Diversity of Plant Terpenes

Plant secondary metabolites involved in environmental adaptation and stress tolerance can be broadly classified into phenolics, flavonoids, alkaloids, and terpenoids. Among these classes of compounds, terpenoids are the most diverse, with significant properties in the context of chemical ecology [1]. These compounds are synthesized from the five-carbon precursor units isopentenyl pyrophosphate (IPP) and its functional isomer dimethylallyl pyrophosphate (DMAPP). Subsequent condensation of IPP and DMAPP through the action of isoprenyl diphosphate synthase (IDS—a type of prenyltransferase (PT)) produces acyclic and achiral isoprenyl diphosphate/pyrophosphate (ID, C_5n_) intermediates including geranyl pyrophosphate (GPP, C_10_), farnesyl pyrophosphate (FPP, C_15_), and geranylgeranyl pyrophosphate (GGPP, C_20_), which are considered universal terpenoid precursors [2]. Terpene synthases (TPSs) act on one or more of these universal precursors (including DMAPP) to produce a combinatorial diversity of terpenes. To clear out any ambiguity, terpenes are simple hydrocarbons based on combinations of DMAPP and ID, while terpenoids (which are also known as isoprenoids) are terpenes with an oxygen moiety and additional structural rearrangements. Nonetheless, these two terms are used interchangeably. Terpenoids are conveniently classified on the basis of the number of carbon atoms they possess: hemiterpenoids (C_5_), monoterpenoids (C_10_), homoterpenoids (C_11,16_), sesquiterpenoids (C_15_), diterpenoids (C_20_), sesterpenoids (C_25_), triterpenoids (C_30_), tetraterpenoids (C_40_), and polyterpenoids (C_>40_, higher-order terpenoids).

A small fraction (1%) of low-molecular-weight (300 Da) compounds known as plant volatiles are implicated in plant–environment interactions and, to some extent, in the abiotic stress response [3,4]. Terpenoids that are emitted to communicate with the environment are referred to as volatile terpenes (VTs) and include hemi-, mono-, homo-, sesqui-, and some diterpenoids. The storage and volatility of these terpenoids are largely influenced by their chemical properties (i.e., vapor pressure and hydrophobicity), while their emission rates and patterns depend on abiotic and biotic factors including temperature, humidity, seasonality, irradiance, and interactions with other plants and organisms [5]. The storage of terpenoids concerns plant tissues with specialized structures such as secretory cavities, resin canals, latex canals, glandular trichomes [6]. A study [5] has emphasized that some VTs, referred to as ‘hidden terpenoids’, are chemically modified by glycosylation for storage and upon exposure to stress, may undergo hydrolysis releasing VTs to the atmosphere on demand. There is a consensus agreement that VT production and emission are tightly associated with stress alleviation, allelopathy, and plant–herbivore–carnivore (tritrophic or multitrophic) interactions.

The chemical diversity of terpenoids is driven by the stereo-specific carbocation cyclization, rearrangement, and elimination reactions that transform a few universal ID precursors into core scaffolds of numerous structurally distinct terpenoids. This indicates that a single TPS can give rise to multiple products due to the stochasticity of bond rearrangement following the generation of the unusual carbocation intermediates. Remarkably, the existence of multi-substrate TPS has also been demonstrated to be dependent on the physiological and development status of plants. This further suggests that substrate preference and terpene product profiles may vary in response to fluctuations in the environment [7]. Functional decoration of these core scaffolds by cytochrome P450 monooxygenases (P450) augments the already-existing chemical diversity through oxygenation and further structural rearrangements, resulting in an estimated 80,000 distinct compounds [8,9].

### 1.2. Stress Response and Terpenoid Biosynthetic Genes

The perception and response of plants to abiotic stresses is a complex process involving multiple genes, proteins, enzymes, and metabolites in a highly regulated network [10,11] (Figure 1). The central component of the salt overly sensitive (SOS) pathway SnRK2 subfamily participates in the terpene phytohormone abscisic acid (ABA) signaling pathway that leads to the activation of transcription factors (TFs) and of the mitogen-activated protein kinase (MAPK) cascade [11,12,13]. Besides membrane-based stress perception, physical stimuli such as thermal stress can be sensed at subcellular locations where disruption of the structure–function relationship of biomolecules occurs.

Given the advancement of next-generation sequencing and its much-reduced cost per base, whole-genome sequencing has become more feasible. The unprecedented chemical diversity of terpenoids is commensurate to the genetic diversity of TPSs found in plant genomes (Figure 2) [14,15,16]. TPS genes were annotated in different numbers, from 29 potentially functional TPS genes in tomato to 69 putatively functional TPS genes in grapevine [17,18]. Enzyme analysis has shown that the emergence of species-specific clades of TPS genes was brought about by neofunctionalization and gene duplication [19]. TPS genes are classified into seven distinct subfamilies/clades, i.e., *TPSa*, *TPSb*, *TPSc*, *TPSd*, *TPSe/f*, *TPSg*, and *TPSh* [20,21]. These subfamilies are mainly discriminated based on the presence and modifications of their N- and C-terminal motifs R(R)X_8_W and D(D)X(X)(D)D, respectively [9]. Phylogenetic analysis of TPS protein families based on PFAM PF01397 and PF03936 illustrates that there is a preferential representation of particular TPS subfamilies (*TPSa*, *TPSb*, *TPSc*, *TPSe/f*, and *TPSg*, Figure 2). Thus far, no genome has been found to host all seven subfamilies of TPS. These TPSs have also undergone lineage-specific expansion wherein each species on average has only 1–2 subfamilies of TPS with probably at least 100 members. Apart from classical plant TPS, microbial TPS-like (MTPSL) exists in non-vascular plants including bryophytes, merchantiophytes, anthocerotophytes, and algae [15].

A recent study has attempted to include *DXS* genes in TPS gene families, suggesting that duplication of DXS and TPS gene families may lead to the separation of biological features in terpenoids production, and overexpression of these genes may result in the production of major component of terpenes [22]. In many plant genomes, *CYP450* and *TPS* gene pairs are juxtaposed right next to each other (Figure 3) [23,24]. Integrated analysis of gene co-expression identified two *CYP450* genes, namely, *CYP88D6* and *CYP72A154*, involved in the triterpenoid saponin glycyrrhizin pathway [25]. In *Arabidopsis*, CYP76C1 acts as linalool metabolizing oxygenase and was found to be co-expressed with TPS10 and TPS14 [26]. In addition to the diverse *TPS* genes, oxidation provided by enzymes of the CYP450 superfamily gives rise to a variety of terpenoid products.

### 1.3. Effects of Plant Terpenes on Abiotic Stress

VTs mitigate the effects of oxidative stress by modulating the oxidative status of plants, and the volatile emission profile could be one of the signature responses of a plant under stress conditions. Considering that there is a convergence of various stress pathways at the level of oxidative signaling [35], it has been proposed [29] that protection against abiotic stress is mediated through direct reactions of terpenoids with oxidants either intracellularly or at the leaf–atmosphere interface/boundary layer, membrane stabilization, and indirect alteration of ROS signaling [35,36]. For instance, thermal stress induces membrane destabilization of the thylakoid membrane, resulting in fragmentation of the membrane and disintegration of protein complexes (e.g., PSII). It is hypothesized that, due to the amphipathic nature of isoprene, they could alleviate stress by transiently inserting in the membrane, which enhances hydrophobic interactions of large protein complexes between themselves or with membrane lipids [37]. On the other hand, membrane-bound antioxidants such as tocopherol and carotenoids (zeaxanthin, neoxanthin, and lutein) may directly scavenge ROS in response to photoinhibition [38,39,40,41,42].

The photosynthetic output is compromised under certain abiotic stresses. Plants have adapted a mechanism to re-focus resources to more important pathways, and it is hypothesized that VTs together with phytohormones are key players in this process called premature or stress-induced senescence [43,44,45]. This is a coping mechanism whereby plants economize available resources. A study [43] has proposed that during abiotic stress, VT production is upregulated by re-routing resources to the MEP pathway and, together with phytohormones, may promote plant senescence and organ abscission. Production of VTs is considered a substantial and irretrievable investment for plants; however, even if photosynthesis is inhibited, VT production and emission are sustained, suggesting they are beneficial to plants under stress conditions [29]. Under one or more abiotic stresses, plants may eventually undergo stress-induced senescence. An article [43] has also proposed that abiotic stress induces VT production and emission, which initiate signaling pathways leading to senescence, apoptosis, and abscission in plants. Senescence in *Arabidopsis* is induced by citral, peppermint, β-pinene, α-pinene, and camphene [43,46,47]. Plants gamble to ensure survival by diverting resources from older organs to synthesize stress-combating secondary metabolites. A correlation has been reported in *Salvia mirzayanni* where increased production of monoterpenoids (1,8-cineole, α-terpinyl acetate, and linalyl acetate) resulted in a subsequent decrease of photosynthetic productivity (rate, stomatal conductance, internal CO_2_ partial pressure, and transpiration) [48]. This observation suggests a resource partitioning that diverts more carbon to secondary metabolite synthesis. Furthermore, an interesting observation during plant senescence of expended leaves is that, while they become more prone to aphid infestation, re-allocation of resources could oppose the ability of aphids to otherwise redirect the flow of photosynthates to them [44]. It could be that re-routing of nutrients and secondary metabolites makes the leaves less deterring to herbivores because of infestation by making it a bountiful source of nutrient leftovers. While this is a compromise that plants have to make, most plants can reverse the process once the environmental conditions become favorable [49]. Abiotic stress-associated terpenes and terpenoids currently known are listed in Table 1.

## 2. Terpenoids in Plant–Insect Interactions

### 2.1. Terpenoids with Toxic and/or Repellent Effects on Insects

Terpenoids are also known to mediate plant–insect (e.g., pollinators, predators, parasitoids, and herbivores) interactions. Generally, the interactions work to the benefit of the plants and to the detriment of herbivores. These interactions are assumed to be mediated by the emission of volatile organic compounds (VOC) including, but not limited to, terpenoids, albeit the fact that a substantial fraction of the VOC consists of terpenoids. The characteristics of VOC mixture are believed depend on the identity and proportion of individual components, and attacking herbivores are sensitive to a particular blends of VOC. The emitted VOC may include one or few with toxic, deterring/repellent, or attractive properties.

Cinnamon and clove essential oils are toxic and repellent to *Sitophilus granaries*, an important grain pest. In addition to these two essential oils, five toxic terpenoids were proposed, i.e., eugenol (considered a terpenoid in that study but not in other works), caryophyllene oxide, α-pinene, α-humulene, and α-phellandrene. Eugenol shows the strongest contact toxicity [58]. It was found that less damage was caused by *Paropsisterna tigrina* adults in *Melalecua alternifolia*, in the presence of a high concentration of terpinolene [59]. In addition, volatiles containing the same terpenoids present in intact and infested rice plants attract or repel the rice pest *Orseolia oryzivora* depending on their concentrations [60] Table 2 summarizes examples from relevant studies.

Unlike terpenoids directly targeting insect pests, terpenoids can act as elicitors to trigger defense mechanisms to pests in plants. Overexpressed β-ocimene from tobacco triggered a high release of methyl salicylate and cis-3-hexen-1-ol in tomato, both of which are proposed to cause impairment of aphid development and reproduction, increasing the defense ability of tomato against aphids [61].

### 2.2. Terpenoids as Attractants to Predators or Parasitoids

Attracting predators or parasitoids is an effective indirect defense adopted by plants against herbivores. *Chrysopa phyllochroma Waesmael* is a promising bio-control candidate as polyphagous predator. It is reported that (Z)-3-hexenyl acetate, (Z)-3-hexenol, (3E)-4,8-dimethyl-1,3,7-nonatriene, and linalool effectively attract *C. phyllochroma* at specific concentrations, while (3E)-4,8-dimethyl-1,3,7-nonatriene and linalool can enhance *C. phyllochroma* female’s ability to parasitoidism [80]. The compound 1,8-cineole, which is a volatile monoterpene cyclic ether [88], can be used as an insecticide. However, the same terpenoid contained in VOCs from cabbage plants attracts the parasitoids *Cotesia glomerata* that lay eggs in *Pieris brassicae*, a caterpillar of specific herbivores. It was reported that the extent of damage caused by these larvae is commensurate to the upregulation of the biosynthesis of the VOC blend [75,76]. Iso-pinocarveol is proposed to potentially act as a recruiter of parasitoids in addition to having toxic properties [86] (Table 2).

Similarly, plant–plant interactions through VTs demonstrates that tomato is induced to release a high level of methyl salicylate and cis-3-hexen-1-ol to attract parasitoids of aphids on exposure to β-ocimene released by neighboring/adjacent tobacco plants [61].

### 2.3. Terpenoids as Attractants to Herbivores

By contrast, unlike those terpenoids potentially involved in the defense against infesting herbivores, some terpenoids act as chemo-attractants to herbivores. This property can be exploited in pest management strategies to intentionally lure herbivores into a trap. Planting transgenic plants overexpressing attractants may effectively lure herbivores into entrapments that prevent escape. This strategy could be particularly useful or practical in mitigating and managing the agricultural and economic impacts of these unwanted insect pests. A study [86] has demonstrated an important association between increased γ-terpinene and α-pinene production in *Eucalyptus grandis* and its subsequent susceptibility to pest infestation, indicating the potential utility of these terpenes as attractants to *E. grandis* pest *Leptocybe invasa*, or a potential decoy/trap when expressed in non-host plant species.

### 2.4. Plant Terpenes with Beneficiaries

Owing to their diversity, terpene products not only provide plants with adaptative strategies for various environmental conditions but also bring benefits especially to long-term cooperators such as pollinators. The above mentioned 1,8-cineole, which can be used as an insecticide, can be tolerated by orchid bees (Euglossinae, e.g., *Eulaema*, *Eufriesea*, *Euglossa*). Male orchid bees are well known fragrance collectors, sensing undiluted 1,8-cineole directly from the flowers of several neotropical orchid genera and species of *Dalechampia* during pollination [62,65]. This monoterpene was reported to be stored in males’ hind leg pouches and later exposed during courtship [89]. On the other hand, female bee species (Apidae: Euglossini), *Hypanthidium* (Megachilidae: Anthidiini), and worker *Trigona* (Apidae: Meliponini) utilize triterpene resins (mixtures of β-amyrin, β-amyrone, dammadienol, and dammadienone) collected from flowers of some *Dalechampia* species as nest construction materials and pollinate these flowers during collection. [90]. While the same resin compound has an inhibitory effect towards generalists and specialized herbivores [91], it is an example of species-specific reward to beneficiaries. Other than providing terpenes as a reward to pollinators, some flowers provide a nursery ground (the flower itself) to the pollinators. Terpene compounds serve as a cue for pollinators to locate their specific host. In the case of the noctuid moths *Hadena bicruris,* these insects are attracted by lilac aldehydes which are the main components of the scent of *Silene latifolia* [67].

Terpene compounds can be finely adjusted to a certain concentration to repel or attract pollinators. Pollination of male and female *Macrozamia lucida* is sort artistic. Thrips (*Cycadothrips chadwick*), the pollinators, are retained in the male cone early in the morning. During midday, after the temperature of the male cone changes, volatile emission increases tremendously (female cone emission is 1/5 of that of male cone). Under such a high concentration of volatile compounds which contain three terpenes (β-myrcene, (E)-β ocimene, and allo-ocimene), the thrips are repelled from the cones. Later, the thrips are attracted to the female cone and finish the whole pollination as they are attracted by a low emission of terpenes, especially β-myrcene [68].

Research studies have also revealed that terpenes serve as signals for pollinators to identify floral stages. In the species-specific relationship between fig (*Ficus hispida*) and wasp (*Ceratosolen solmsi marchali*), the pollinator needs to identify the female floral stage because it is the only stage receptive to pollinators. Three monoterpenes, namely, linalool, limonene, and β-pinene, can be found in receptive fig volatiles. Wasps are responsive to both quantity (change in concentration) and quality (substitution with different isomers), but quality does matter regarding locating the host and floral stage differentiation [69].

*Peperomia macrostachya* produces seeds that attract arboreal in nutrient-rich formicaries. This species-specific symbiosis is called neotropical ant-gardens (AGs). The seeds emit numerous phenolic and terpenoid volatiles including geranyl linalool. This terpenoid can be perceived by the seed-dispersal partner AG ant species *Camponotus femoratus*; as a result, these ants are attracted to the location of the scent. This scent is rather toxic to most insect species [73]. This explains why this plant is distributed largely around AG ant nests [74].

## 3. Potential Agricultural Biotechnology Applications

Terpenoids play vital roles in different biological processes, not only for plant defenses but also for insect development. Sesquiterpenoids of insects include the well-known farnesoic acid (FA), methyl farnesoate (MF), and juvenile hormone (JH) [19,92,93]. The production of sesquiterpenoids in both plants and insects starts with the formation of the C5 building units IPP and DMAPP by the mevalonate (MVA) pathway and the alternate plastidial methylerythritol phosphate (MEP) pathway in plants. The C5 building units are further assembled into farnesyl diphosphate (FPP) and modified into C15 sesquiterpenoid products with the help of different TPSs [19,94,95,96]. Yet, diversity can be found in the production of sesquiterpenoids. For instance, different groups of insects use different pathways to produce JH. While most insects like cockroaches and locust utilize epoxidase CYP15A1 to produce JH, lepidopterans first convert FA to JH-III acid with epoxidase CYP15C1 and subsequently use juvenile hormone acid O-methyltransferase (JHAMT) to form JH [92,96,97,98]. Besides its role in regulating various endogenous processes, JH is also involved in mediating insect responses to the environment. For instance, in response to seasonal changes of the photoperiod, aphids switch their reproductive mode between parthenogenesis and sexual reproduction, during which, JH appears to have a role, because transcriptome analysis found that genes (JH acid methyltransferase, JH binding protein-like, circadian clock-controlled protein) responsible for JH synthesis and transport are down-regulated when days are short [99]. Despite the similar chemical structures of terpenoids in plants and sesquiterpenoids in insects, there are currently no studies showing a direct crosstalk between the two kingdoms.

Nevertheless, terpenoids that attract herbivores could also function as bioherbicides allowing management of weed growth in paddies or any agricultural fields. *Altica cyanea* females are attracted by elevated emissions of the volatile blend from damaged *Ludwigia octovalvis* (Jacq.) Raven (Onagraceae), a rice-field weed, in which four terpenoids were identified, i.e., α-pinene, linalool oxide, geraniol, and phytol [85]. To keep pests away, about 20 years ago a project proposed to transfer the (S)-limonene synthase gene from wildflower *Clarkia breweri* to petunia. Linalool, which can repel aphids, successfully increased [100]. A similar overexpression of linalool synthase gene from strawberry in chrysanthemum also decreases the tendency of thrips and aphids to attack the transgenic plants with respect control plants [101]. Metabolic engineering involved the strawberry *FaNES1* (*Fragaria ananassa* Nerolidol Synthase 1) gene expressed in other crops such as potato. FaNES1 recombinant protein catalyzes the formation of S-linalool, resulting in a much higher level of linalool [102]. Interestingly, targeting the FaNES1 protein in the mitochondria of transgenic *Arabidopsis* rather than the protein in plastid can lead to a higher yield of the sesquiterpene nerolidol [103] The homoterpene derivative of nerolidol known as 4,8-dimethyl-1,3(E),7-nonatriene further contributes to plant defense by attracting insect herbivores predators [104]. Another indirect defense example of maize, listed in the above Table 2, consisted in inducing the sesquiterpene synthase gene (*TPS10*) from maize into *Arabidopsis* to help the female parasitoid wasp *C. marginiventris* to locate the host caterpillars [105]. Genetically modified crops can be more attractive to pollinators due to their richness in terpene compounds. Comparing Cry3Bb-expressing genetically modified (GM) eggplants and their near-isogenic control, the GM eggplant contains an abundant amount of terpenes, such as (+)-limonene, Z-jasmone, p-cymene, α-pinene, and (−)-limonene, which explains the increased visitation by the commercial pollinator bumblebees (*Bombus terrestris*) [106]. With nanotechnology, encapsulated terpenoids can be used to trap insect pests. A study [87] evaluated and proposed a potential system to trap *Bemisia tabaci* through attraction by geraniol encapsulated in chitosan/gum Arabic nanoparticles.

Other than promoting crop yield through weed control, pest management, and pollination, one of the direct impacts on agriculture by mustering terpenoid biosynthesis is to enhance or alter the scents and flavors of products. There are cases of increased production of monoterpenes in transgenic plants such as tomato, carnation, potato, mint, and tobacco by overexpressing the corresponding *TPS* genes or *CYP450* genes or regulating TFs, promoters, etc. [107,108,109,110]. The same goes for medicinal plants. A sesquiterpene (artemisinin) produced by *Artemisia annua* is regarded as an effective therapy against malaria-causing *Plasmodium falciparum* strains [111]. Artemisinin biosynthesis can be upregulated via two TFs (AaERF1 and AaERF2) belonging to the AP2/ERF family [112]. More engineering approaches such as suppressing the metabolic pathways that compete with the biosynthetic pathways of terpenoid or regulating endogenous phytohormones can be adopted to adjust terpenoid production [113] in response to the challenging agriculture environment.

Abiotic stresses including drought, flooding, changes in salinity, heat, cold, and high irradiance have caused a decline in plant primary production and crop losses worldwide. Hence, mitigating the negative impacts of these factors is imperative to address pressing issues regarding food quality and security. Considering the important roles of terpenoids in the abiotic stress response, metabolic engineering towards a more stable and efficient production of volatile and essential terpenoids may provide a long-term solution. Metabolic engineering of plant terpenoids has been reviewed [114], and while increasing the production of terpenoids is the desired ultimate goal, it comes with a cost, particularly because of the possibility of perturbing cellular homeostasis. Thus, it is equally important to consider a different paradigm in this matter. A preferential increase in precursor flux in one pathway might inadvertently deplete other pathways with shared precursors needed for the production of equally essential metabolites, and unregulated production of volatile terpenes may lead to cytotoxicity. In retrospect, the key role of phytohormones in plant defense and stress responses is widely recognized. These hormones activate stress signaling in a global manner, culminating with an integrated stress response including, but not limited to, increased production of volatile terpenes [115]. A proportionate regulation of biochemical pathways in response to changing environmental conditions is intuitively less stressful and perturbing to plants that are already experiencing stress. Hence, modifying the production of stress-associated phytohormones and modulating their potentiating effect on the stress response may provide a more suitable alternative to increasing volatile terpene production, while keeping any adverse effect that might spontaneously arise at bay.

## 4. Conclusions

The interactions between plants, environment, and insects are inevitable. Secondary metabolites, particularly the emitted volatile organic compounds and volatile terpenes, are implicated in abiotic and biotic stress tolerance. Terpenoids, in particular, have key roles in alleviating abiotic stresses, in mitigating herbivore infestation, and in mediating communications with third-party beneficial species such as those involved in tritrophic and multitrophic interactions. Plant–insect interactions represent a perpetual arms race, a continuous process of out-smarting one another. This process has resulted into an unprecedented diversity of terpenes and terpenoids in both plants and insects. How to get a thorough understanding of their dynamic interactions and how to make an appropriate use of them in biotechnology are questions that will need to be tackled in this coming decade.

## Figures and Tables

**Figure 1 ijms-21-07382-f001:**
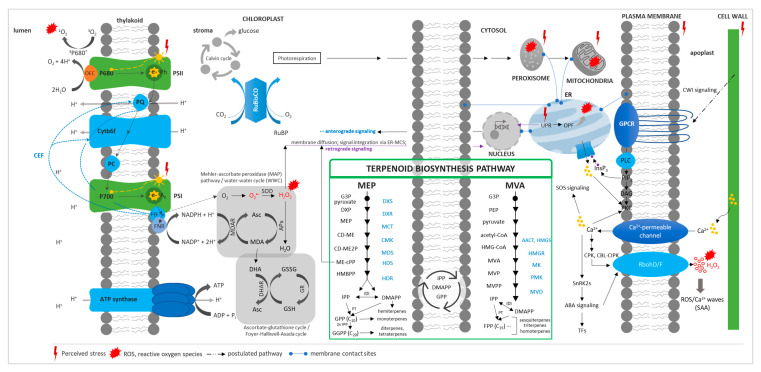
Plant abiotic stress perception and response, concerning different cell compartments/organelles, and the terpenoid biosynthesis pathway. Stress perception is sensed anywhere in the cell where there is a perturbation of biomolecules and metabolic reactions in various cellular compartments/organelles. Ca^2+^ signaling is initiated through [Ca^2+^]_cyt_ changes due to the influx of Ca^2+^ or its release from intracellular calcium stores. Changes in [Ca^2+^]_cyt_ are sensed by various calcium-binding proteins (e.g., calmodulin, CBL), and signals are relayed to the nucleus through the SOS pathway. Ca^2+^ also activates other pathways such as ABA signaling. Reactive oxygen species (ROS) production in the apoplast is initiated in response to Ca^2+^ and ABA signaling, and, together, Ca^2+^/ROS signals are automatically propagated to initiate an SAA or distal response. Mechano-sensitive channels through the CWI pathway may also play a role in sensing cellular stress and initiating a signal transduction cascade. ROS during stress is generated at various hotspots, particularly in metabolically active organelles, i.e., chloroplast, mitochondria, peroxisomes, and the endoplasmic reticulum. When ROS production exceeds the quenching capacity of the cell or cellular organelles, retrograde signaling is initiated either through ROS or through ROS-activated/dependent signaling pathways. ROS signals from different organelles may be integrated and fine-tuned through the ER–MCS. To prevent photoinhibition in the chloroplast, particularly that of PSII, CEF, Mehler peroxidase reaction, and photorespiration may relieve the system of the excess of reducing equivalents, while antioxidant defense systems such as the ascorbate–glutathione cycle may alleviate oxidative stress by quenching ROS. PSI/II, photosystem I/II; Ph, pheophytin; PQ, plastoquinone; Cytb6f, cytochrome b6/f complex; FD, ferredoxin; FNR; ferredoxin NADP^+^ reductase; CEF, cyclic electron flow; RuBP, ribulose-1,5-biphosphate; RuBisCO, ribulose-1,5-biphosphate carboxylase/oxygenase; SOD, superoxide dismutase; Asc, ascorbate; MDA, monodehydroascorbate; APx, ascorbate peroxidase; MDAR, monodehydroascorbate reductase; DHA, dehydroascorbate; GSSG, oxidized glutathione; GSH, reduced glutathione; GR, glutathione reductase; DHAR, dehydroascorbate reductase; SAA, systemic acquired acclimation; CWI, cell-wall integrity; GPCR, G-protein-coupled receptors; PLC, phospholipase C; PIP_2_, phosphatidylinositol-4,5-biphosphate; DAG, diacylglycerol; PKC, protein kinase C; InsP_3_, inositol-1,4,5-triphosphate; CPK, calcium-dependent protein kinase; CBL, calcineurin B-like; CIPK, CBL-interacting protein kinase; SOS, salt overly sensitive; SnRK, SNF1/AMPK-related kinase; ABA, abscisic acid; RbohD/F, respiratory burst oxidase homolog D/F; TFs, transcription factors; ER, endoplasmic reticulum; ER–MCS, ER–membrane contact sites; UPR, unfolded protein response; OPF, oxidative protein folding; MEP, 2-C-methyl-D-erythritol-4-phosphate; MVA, mevalonic acid; G3P, glyceraldehyde-3-phosphate; DXS, 1-deoxy-D-xylulose-5-phosphate synthase; DXP, 1-deoxy-D-xylulose-5-phosphate; DXR, 1-deoxy-D-xylulose-5-phosphate reductoisomerase; MCT; 2-C-methyl-d-erythritol-4-phosphate cytidylyltransferase; CD-ME, 4-(cytidine-5′-diphospho)-2-C-methyl-D-erythritol; CMK, 4-diphosphocytidyl-2-C-methyl-D-erythritol kinase, CD-ME2P, 4-(cytidine-5′-diphospho)-2-C-methyl-D-erythritol-2-phosphate; MDS, 2-C-methyl-d-erythritol-2,4-cyclodiphosphate synthase; ME-cPP, methylerythritol cyclodiphosphate; HDS, 4-hydroxy-3-methylbut-2-enyl- diphosphate synthase; PEP, phosphoenolpyruvate; AACT, acetyl-CoA acetyltransferase; HMGS, HMG-CoA synthase; HMGR, HMG-CoA reductase; MVA, mevalonate/mevalonic acid; MK, mevalonate kinase; MVP, mevalonate-5-phosphate; PMK, phosphomevalonate kinase; MVPP, mevalonate pyrpophosphate; MVD, diphosphomevalonate decarboxylase; IPP, isopentenyl pyrophosphate; DMAPP, dimethylallyl pyrophosphate; IDI, IPP-DMAPP isomerase; PT, prenyl transferase; GPP, geranyl pyrophosphate; FPP, farnesyl pyrophosphate; GGPP, geranylgeranyl pyrophosphate. Photosynthetic REDOX signals and quenching are an adaptation of [27], while the terpenoid biosynthesis pathway is directly adopted from [28,29].

**Figure 2 ijms-21-07382-f002:**
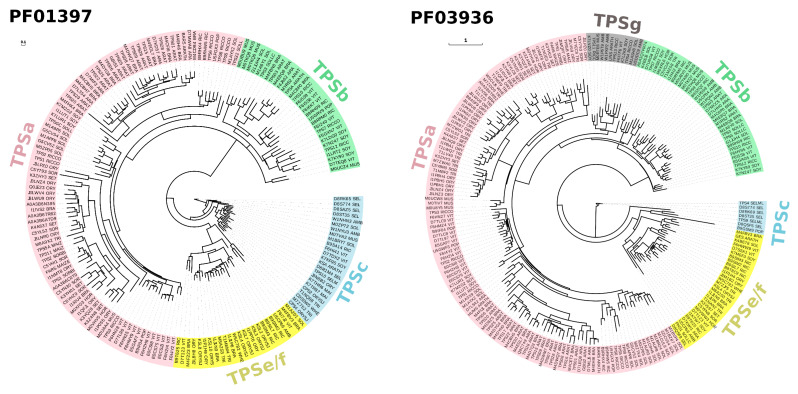
Phylogenetic classification of terpene synthases (TPSs) based on PFAM PF01397 and PF03936 SEED alignments reconstructed using PhyML 3.0 maximum likelihood method [30,31]. Automatic model selection was based on Smart Model Selection (SMS) in PhyML [32], while the aLRT SH-like algorithm was used in place of bootstrap analysis. TPS protein sequences were classified into subfamilies using Terzyme [33]. The resulting phylogenetic trees were visualized using Evolview v3 [34]. PFAM Accession numbers/IDs are indicated at the leaf nodes, with corresponding background colors representing *TPS* gene subfamily affiliations. Bars (0.1 and 1) indicate mean amino acid substitution per site.

**Figure 3 ijms-21-07382-f003:**
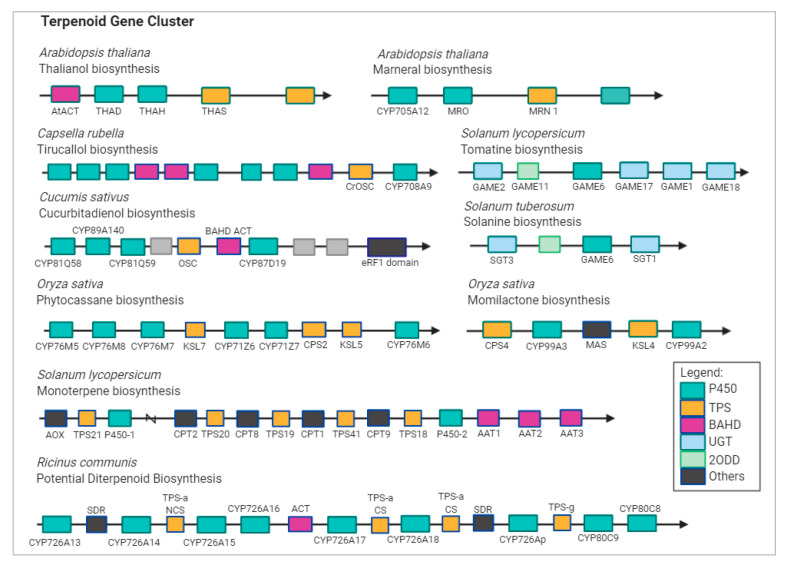
Terpenoid biosynthetic gene cluster in plant genomes. Abbreviations: AAT, alcohol acyl transferases; ACT, acyltransferase; AOX, aldehyde oxidase; CPS, labdadienyl/copalyl synthases; CS, casbene synthase; KSL, kaurene synthase-like gene; MAS, momilactone A synthase; THAH, thalianol hydroxylase; THAS, thalianol synthase; MRO, putative marneral oxidase; MRN, marneral synthase; OSC, oxidosqualene synthase; GAME, glycoalkaloid metabolism; SGT, sterol alkaloid glycosyltransferase; TPS, terpene synthase; P450, CYP; NCS, neo-cembrene synthase; SDR, short-chain alcohol dehydrogenase/reductase; 2ODD, 2-oxoglutarate-dependent dioxygenase; UGT, UDP-sugar dependent glycosyltransferase family.

**Table 1 ijms-21-07382-t001:** Abiotic stress-associated terpenes and terpenoids.

Plant Species	Terpenes	Abiotic Stress	References
***Nicotiana*** ***attenuata***	sesquiterpenes: *(E)*-β-Farnesene; *(E)*-α-Bergamotene	Tropospheric ozone, ROS	[50,51,52]
***Oryza*** ***sativa***	monoterpenes: limonene; sabinene, myrcene; α-terpinene; β-ocimene; γ-terpinene; α-terpinolene	High irradiance (UV-B, H_2_O_2_, and γ rays)	[53]
***Vitis vinifera*** **cv. Chardonay**	monoterpenes	Thermal stress	[54]
***Pseudotsuga*** ***menziesii* (Douglas fir)**	monoterpenes: β-pinene; α-pinene; β-citronellol; 3-carene; camphene	Drought- and salt-induced stress	[55]
***Zea mays***	Sesquiterpenoid: zealexin; diterpenoid: kaulalexins together with ABA		[56]
***Salvia*** ***officinalis, Salvia fruticose, Rosmarinus officinalis***	diterpene: carnosic acid		[57]

**Table 2 ijms-21-07382-t002:** Plant terpenes and their observed roles and effects against insects.

Plant Species	Terpenes	Targets	Effects	References
Tropical orchids	monoterpene: 1, 8-cineole	Male euglossine bees	an attractant and reward to pollinator	[62]
Dalechampia (Euphorbiaceae) epiphytes (*Clusia*)	terpenoid resins (oxygenated triterpenes)	Female euglossine (Apidae), female anthidiine (Megachilidae), or worker meliponine (Apidae) bees	reward to pollinators for use in nest construction	[63,64,65]
Kiwifruit (*Actinidia deliciosa*)	Sesquiterpene: α-farnesene, germacrene D, Monoterpenes: (*E*)-β-ocimene, (*Z*,*E*)-α-farnesene	Mainly honeybees (Apidae)	attract a variety of pollinators	[66]
White Campion *(Silene latifolia)*	lilac aldehydes	Noctuid moths (*Hadena bicruris)*	scent cue for pollinators to locate their specific host	[67]
Pineapple zamia (*Macrozamia lucida*)	β-myrcene, (E)-β ocimene and allo-ocimene	Thrips (*Cycadothrips chadwick*)	repel or attract pollinators to complete the pollination from male to female	[68]
Fig (*Ficus hispida*)	Monoterpenes: linalool, limonene and β-pinene	Wasp (*Ceratosolen solmsi marchali*)	signals for pollinators to identify floral stages	[69]
Monkeyflower (*Mimulus lewisii*)	D-limonene, β-myrcene and E-β-ocimene.	Bumblebee (Bombus vosnesenskii)	attract specific pollinators	[70]
Sweet rocket *(Hesperis Matronalis)*	linalool, β-ocimene.	Mainly syrphid flies (Syrphidae)	attract a variety of pollinators	[71,72]
Radiator plant *(Peperomia macrostachya)*	geranyl linalool	Arboreal ants *(Camponotus femoratus)*	attract seed disperser to collect and plant their seeds	[73,74]
Cabbage plants (*Brassica*)	monoterpene: 1,8-cineole	Parasitic wasps (*Cotesia glomerata)*	attracts parasitoids that lay eggs in the caterpillars of specific herbivores	[75,76]
Maize (*Zea mays*)	terpene	Parasitic wasps (*Cotesia marginventris*) specialized parasitoid (*Microplitis croceipes*).	attracts the endoparasitoid that attacks larvae of a wide range of lepidopterous hosts	[77]
Elm *(Ulmus minor)*	homoterpenoids: (E)-2,6-dimethyl-6,8-nonadien-4-one, (E)-2,6-dimethyl-2,6,8-nonatrien-4-one, and (R,E)-2,3-epoxy-2,6-dimethyl-6,8-nonadiene	Parasitoid (*Oomyzus gallerucae)*	attracts egg parasitoids to attack the eggs of elm leaf beetles *Xanthogaleruca luteola*	[78,79]
Various plant species	(Z)-3-hexenyl acetate, (Z)-3-hexenol, (3E)-4,8-dimethyl-1,3,7-nonatriene, and linalool	Pest predator *(Chrysopa phyllochroma)*	specific concentrations of these terpenes are significantly attractive to this target	[80]
Various plant species	(Z)-3-hexenyl acetate, (3E)-4,8-dimethyl-1,3,7-nonatriene, and linalool	Pest predator *(Chrysopa phyllochroma)*	promote oviposition	[80]
Tomato and tobacco	β-ocimene	Parasitoid (*Aphidius ervi)*	attract parasitoids	[61]
Tomato and tobacco	β-ocimene	Pest (*Macrosiphum euphorbiae)*	defense against pest	[61]
*Melalecua alternifolia*	Terpinolene	*Paropsisterna tigrina*	pest adults cause less damage in the presence of a high level of terpinolene in the plant	[59]
Brazil nut family (Lecythidaceae)	S-methylmethionine,	Wood-boring longicorn beetles (Cerambycidae)	deterrent to specialist beetle seeking oviposition sites	[81]
Lavender (*Lavandula angustifolia*)	β-trans-ocimene, (+)-R-limonene	Aphids	deterrent to pest	[82]
Cucumber (*Cucumis sativus*)	Tetracyclic terpenes: Cucurbitacins	Spider mite (*Tetranychus urticae*)	antibiotic effect on spider mites but attractive to the pest cucumber beetle	[83,84]
Cinnamon and clove	Eugenol, caryophyllene oxide, α-pinene, α-humulene and α-phellandrene	*Sitophilus granarius*	toxic and repellent effects to adult pest	[58]
Water primrose (*Ludwigia octovalvis*)	α-pinene, linalool oxide, geraniol, and phytol	Weber (*Altica cyanea*)	attractive to pest females	[85]
Rice (*Oryza sativa*).	(S)-linalool, 4,8-dimethyl-1,3,7-nonatriene, (E)-caryophyllene, and (R/S)-(E)-nerolidol	African rice gall midge (*Orseolia oryzivora*)	attractive to mated female pest in intact rice, but repellent with different concentrations of the same volatiles in infested plant	[60]
*Eucalyptus grandis*	α-pinene, γ-terpinene	*Leptocybe invasa*	potentially attractive to pest	[86]
Various plant species	Geraniol	*Bemisia tabaci*	encapsulated geraniol shows attraction to *B. tabaci*	[87]

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
