# Peer review of "Terpenes and Terpenoids in Plants: Interactions with Environment and Insects"

_ijms, 2020, doi:10.3390/ijms21197382_

Round 1

Reviewer 1 Report

The review manuscript entitled "Terpenes and terpenoids in plants: Interactions with environment and insects" by H-M. Lam, T-F. Chan, J. H.L. Hui et al. detail an overview of plant-environment and plant-insect interactions in the context of terpenes and terpenoids serving as important chemical mediators of these abiotic and biotic interactions. The overall the manuscript seemed well organized. Although some abbreviations are unfamiliar (e.g. CWI) and the whole manuscript would need minor revision, this review was considered to be worth accepted for publication.

Examples of specific comments on editorial matter:

p2. Line 81, 90-91; p3 line101; p10 line 64; p11 line 84; 

Error! Reference source not found.

These need to be corrected

p9, Line 44: are listed in Error! Reference source not found.

It should be corrected to certain table

Reference section:

format needs to be revised

Author Response

We thank the reviewer for the positive and constructive comments. The suggestions have all been incorporated in the latest version of the manuscript.

Reviewer 2 Report

The review describes how terpenes/terpenoids interact with insects under selective pressure.

The manuscript is well written - in the version I had there were some issue with references( incorrectly formatted ) which needs to be fixed. 

The merits of this review is that it addresses the co-evolution between terpenes/secondary metabolites in plants and their environment. The significance of the article is decent, meaning that it summarizes the findings in the field and gives the reader a good insight into the field. The narrative is presented in a sound way and is very precise in its description. It gives a nice overview of the field and how little is known at especially the molecular levels of these dependencies.

Besides, what I have already addressed is the formatting of the references which needs to be fixed. Some minor language formulations, especially in the abstract, could be improved.

p2. l: 80,90
p3 l:101
p9 l44
p10 l64
p11 l84

So accepted with minor revision after fixing especially the formatting.

L1 of the abstract is a bit hard to grasp.

Otherwise fine. 

Author Response

(The authors gave the same response as above.)
